# Generating Component Designs for an Improved NVH Performance by Using an Artificial Neural Network as an Optimization Metamodel

**Timo von Wysocki** [1,2,*] ⬦, **Frank Rieger** [1,2] ⬦, **Dimitrios Ernst Tsokaktsidis** [2,3] ⬦ and **Frank Gauterin** [1] ⬦

1  Institute of Vehicle System Technology, Karlsruhe Institute of Technology, 76131 Karlsruhe, Germany; frankrieger.94@gmail.com (F.R.); frank.gauterin@kit.edu (F.G.)
2  Mercedes-Benz AG, 70372 Stuttgart, Germany; dimitrios_ernst.tsokaktsidis@daimler.com
3  Chair of Vibro-Acoustics of Vehicles and Machines, Technical University of Munich, 85748 Garching, Germany
*  Correspondence: timo.wysocki@kit.edu

**Abstract:** In modern vehicle development, suspension components have to meet many boundary conditions. In noise, vibration, and harshness (NVH) development these are for example eigenfrequencies and frequency response function (FRF) amplitudes. Component geometry parameters, for example kinematic hard points, often affect multiple of these targets in a non intuitive way. In this article, we present a practical approach to find optimized parameters for a component design, which fulfills an FRF target curve. By morphing an initial component finite element model we create training data for an artificial neural network (ANN) which predicts FRFs from geometry parameter input. Then the ANN serves as a metamodel for an evolutionary algorithm optimizer which identifies fitting geometry parameter sets, meeting an FRF target curve. The methodology enables a component design which considers an FRF as a component target. In multiple simulation examples we demonstrate the capability of identifying component designs modifying specific eigenfrequency or amplitude features of the FRFs.

**Keywords:** component design; optimization; artificial neural network; morphing; FEM; vibration; acoustics; NVH; boundary conditions; simulation





## 1. Introduction

Road noise is currently becoming an even more relevant topic for modern vehicles because electrification and quieter engine noise in general make road noise more dominant for the passengers. As road noise is annoying for the passengers, its reduction is an important task in the vehicle development process [1–5].

Road noise, transmitted from the tire contact patch through the suspension into the passenger cabin, makes the suspension transfer path one of the main assembly groups focused in the noise, vibration, and harshness (NVH) development. Though, the design of suspension components is subject to many different, often conflicting boundary conditions and optimization targets from different development domains [6,7]. For example, in the domain of NVH, a component could face a target for two different eigenfrequencies, since they must not match the eigenfrequency of a neighboring component in the same transfer path [3]. If one of the eigenfrequencies already fulfills the target criterion, the other one has to be modified without changing the first one.

The transfer path through the suspension system is characterized by a frequency response function (FRF) which correlates the input and the output of the transfer path. Noise transfer in general can be reduced either by reducing the excitation at the source or by reducing the transmission through the transfer path [6,8]. Reduction in the noise transfer in the transfer path can be achieved either by changing the component design or

by implementing active systems [5]. As passive transfer paths still dominate suspension technology [9–11], we focus on a reduction in the passive transfer path by changing the position of kinematic hard points. These are the connection points between the individual suspension components and determine the kinematic characteristic of the suspension. Earlier publications suggested kinematics modifications as an opportunity to improve interior road noise in the early development phase [12–14]. The presented methodology is a practical approach to identify kinematics modifications suitable for such noise reduction.

As considering different development requirements becomes more difficult in later development phases, car manufacturers focus on a noise reduction in the early development phase [3], when few geometry parameters define the shape and properties of a component. In this digital design phase, no hardware is available and suspension concepts often change [15,16]. In the volatile development environment, simulation methods help finding and prioritizing possible design variants. Finding a global optimum for only one requirement is not the major target, whereas generating designs that fulfill multiple requirements is desirable [17–19].

In the early digital development phase, noise transfer development usually relies on finite element (FE) simulation [20]. In opposition to a topology optimization, in this article we focus on a geometry parameter optimization. Here, few parameters describe the geometric design of the component. Often, the geometry optimization works directly on the FE model, which requires high computation times. In a holistic and fast development process the optimization should be based on fast calculating metamodels that represent the behavior of the FE model [13,18,21–26]. In a previous publication, we showed polynomial metamodels to be capable of a targeted reduction in specific frequency bands of a component FRF [27]. Polynomial metamodels perform well for simple, continuous, and differentiable phenomena (p. 48, ref. [25]). If there are many parameters and complex systems like full vehicles, these constraints cannot be satisfied (p. 106, ref. [13]). In the automotive industry, artificial neural networks (ANN) currently establish themselves as a solution to overcome these obstacles [13,21,24,26,28,29]. In a previous publication we used an ANN to predict the FRF of an abstract component based on one geometry parameter [30]. We showed the network's capability of predicting the FRF of unseen geometry configurations. In this article, we use the ANN as a metamodel in an optimization process which creates geometry parameter sets for a real suspension component, which fulfills an FRF target curve.

We use a data set of 500 different component designs and their respective FRFs to train ANNs which predict these FRFs from the geometry parameter sets. A component of a vehicle suspension is used to demonstrate the capabilities. We specify different FRF target curves and use the optimizer to create fitting geometry parameter sets.

## 2. Approach

In the following sections we present the approach used in this article, including the use case scenario and the methodology.

### 2.1. Use Case Scenario

The component under investigation is the knuckle of a vehicle suspension. The same knuckle was also used as an example in a previous publication [27].

Figure 1 is a simplified representation of the knuckle. The flat reddish flange surface represents the wheel hub attachment surface. We simplified the flange surface by a massless rigid element into a single wheel center point. This point is marked by a sphere in the center of the flange surface. The gray surfaces are attachment surfaces to suspension links. A massless rigid element combines each of the attachment surfaces into one center point for force examination, too. These points are represented by gray spheres in the center of each attachment surface. We call these points kinematic hard points. In Figure 1, all five hard points including their attachment surfaces and the flange surface including the wheel center point are visualized. In the previous publication [27], we only used the coordinate

of the hard point for the track rod (labeled $P$ in Figure 1) as a variable geometry parameter. In this article, we use all 5 hard point coordinates. Each hard point coordinate $x_i$

$$x_i = [x_{i,1}, x_{i,2}, x_{i,3}]^\mathsf{T} \tag{1}$$

consists of three direction components in the coordinate system presented in Figure 1. For the 5 hard points this results in 15 variable geometry parameters—the 15 design variables. In the scenario for this article the geometry parameters represent a suspension kinematics modification. This is a common demand in the early digital vehicle development phase.

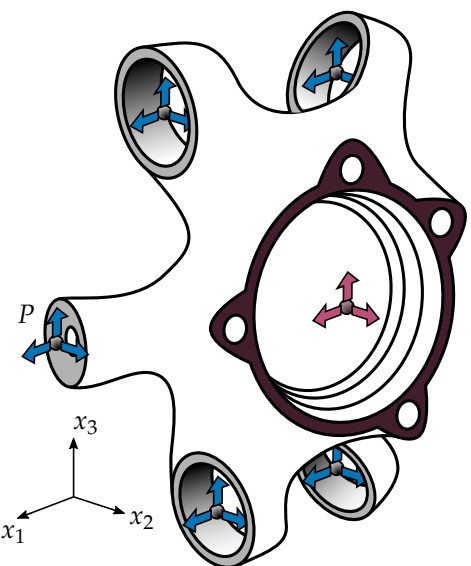

**Figure 1.** Knuckle of a vehicle suspension. The flat reddish surface in the center represents the wheel hub attachment surface. Gray spheres inside gray surfaces represent variable kinematic hard points. FRFs describe the noise transfer between each blue arrow and each red arrow. Each arrow represents one of the three directions $x_1$, $x_2$ or $x_3$.

For the investigation of the noise transfer we calculate FRFs for the component. We use forces at the input and the output of the transfer path. This results in a non-dimensional FRF. Because the component is fixed to ground via stiff springs, the used FRF is directly proportional to the admittance, which correlates response displacements to excitation forces [31]. We examine the FRFs between the 5 hard points ($p = 5$) and the wheel center, marked by five gray and one reddish spheres in Figure 1. Each arrow represents one of the three directions $x_1$, $x_2$ or $x_3$. There are FRFs from each excitation direction ($q = 3$) to each receiver direction ($r = 3$) resulting in

$$n = p \cdot q \cdot r \tag{2}$$
$$= 5 \cdot 3 \cdot 3 = 45 \tag{3}$$

FRFs for the component. Each FRF covers the frequency range $0\,\text{Hz}$ to $2000\,\text{Hz}$. Since we use discrete FE simulation, each FRF consists of discrete steps, called frequency bins, instead of a continuous function. The used frequency step width is $2\,\text{Hz}$ resulting in 1000 frequency bins per FRF.

*2.2. Workflow*

The tool chain used for the optimization is presented in Figure 2.

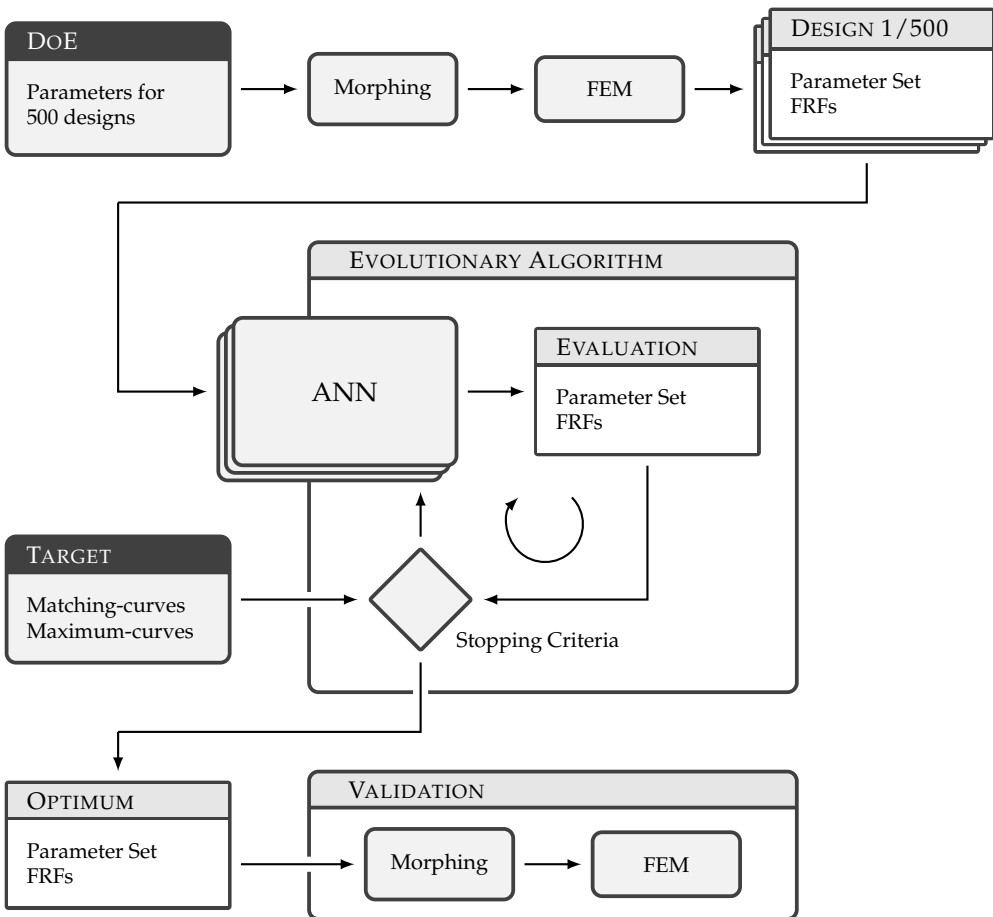

**Figure 2.** Flow chart representing the presented approach. The simulated designs created via DoE serve as training data for multiple ANNs. The optimizer identifies fitting parameter sets that fulfill the target curves. The identified design is then validated.

Using a space filling latin hypercube (SFLH) algorithm as a design of experiments (DoE) method, we create a data set for the 15 variable geometry parameters. The 15 parameters are allowed to move up to ±10 mm independently from each other. This represents a reasonable order of magnitude for kinematics modifications in the suspension development process. We use Ansys optiSLang [32] to create the DoE parameter sets containing 500 designs. In the actual development process there are multiple additional scenarios besides the scenario presented in this article. As we need these different scenarios to be based on the same DoE and require fast component revision times, 500 samples offer an acceptable trade-off between required simulation time and metamodel accuracy.

For each of the 500 parameter sets we automatically create an FE model of the knuckle. This is performed by morphing an initial component FE model using the free form morphing algorithm presented in earlier publications [14,27]. In one of them, we present an example of the FE mesh [27]. The initial component FE model represents the metal part of the component and consists of approximately 100,000 FE nodes and 50,000 FE elements.

In order to guarantee adequate mesh quality after the morphing process, we perform a mesh quality analysis. We assume a worst case scenario in which all 15 geometry parameters are morphed with the maximum displacement of 10 mm. Then we compare the mesh quality of the morphed component FE model to the initial component FE model. As the initial component FE model is a well validated simulation model used in the vehicle development process, the mesh quality after morphing should not be decreased. We compare the element aspect ratio and the element skewness. We identify the worst element for both criteria in the initial component FE model. For the worst case scenario, no

element transforms to a worse aspect ratio. Only 2 of the 50,000 FE elements exceed the original skewness. This indicates a similar mesh quality to the initial component FE model.

The FE models are solved using *NX Nastran* with *SOL 111* and provide the 45 FRFs for each of them. This data set of 45 FRFs for 500 parameter sets of 15 variable parameters is the basis for all further investigations.

The data set provides training data for 45 ANNs. Each FRF is represented by a unique ANN. The ANN setup is described in Section 2.3. The input for each of the ANNs are the 15 geometry parameters and the output are the frequency bins of one of the 45 FRFs.

The ANNs serve as metamodels for the optimizer. Here, we use an evolutionary algorithm (EA). The EA cycles through the creation of parameter sets for new designs and uses the ANNs to predict the FRFs for them. The design space for the EA is restricted to the DoE space at most, as extrapolating from metamodels is usually prohibited (p. 24, ref. [25]).

The ANNs could also have been trained to output the optimum parameter set regarding the target curves by using the internal optimizer of the ANN while training. We chose to output the FRFs, because we want to be able to change optimization criteria without retraining the net and perform multi-objective optimizations with multiple FRFs. Furthermore, we want to receive multiple parameter set suggestions in order to direct the design process, rather than finding the one optimum design.

To evaluate stopping criteria, the EA compares the respective FRF to the given target curve. The target curve is a desired curve for one of the 45 component FRFs. This could be a matching-curve, which is a target FRF to be matched as precisely as possible, or a maximum-curve, which must not be exceeded at any frequency bin. Section 2.4 describes the creation method for the target curve.

If the optimizer identifies one or multiple optimum parameter sets, it hands them over to the validation. For each parameter set, we create a validation FE model by morphing the initial component FE model. The simulated FRF is then compared to the predicted one, in order to approve the identified designs.

### 2.3. Artificial Neural Network

For the optimization metamodels, we use ANNs. Given non-linear activations, ANNs are universal approximators [33]. Without explicitly choosing an approach function they are able to extract relations from a presented parameter set. In our case, we use one densely connected ANN according to Table 1 for each of the 45 FRFs.

**Table 1.** Architecture for each trained ANN.

| Layer | Size | Activation |
|---|---|---|
| In Layer $d_{\mathrm{IN}}$ | 15 | – |
| Layer 1 $d_{\mathrm{HL1}}$ | $2^{10}$ | ReLU |
| Layer 2 $d_{\mathrm{HL2}}$ | $2^{10}$ | ReLU |
| Layer 3 $d_{\mathrm{HL3}}$ | $2^{10}$ | ReLU |
| Out Layer $d_{\mathrm{OUT}}$ | 1000 | Identity |

The input layer dimension $d_{\mathrm{IN}}$ is given by the 15 geometry parameters explained in Equation (1). The hidden layers (HL) have the size $d_{\mathrm{HL1}}$ through $d_{\mathrm{HL3}}$. These were obtained empirically by testing varying layer sizes and observing the corresponding validation loss. The layer sizes $d_{\mathrm{HL1}}$ through $d_{\mathrm{HL3}}$ represent the configuration which minimizes the validation loss. Thus, the presented ANNs reflect the appropriate complexity.

Each hidden layer is applied with rectified linear unit (ReLU) activations to enable non-linear mapping. The output layer size $d_{\mathrm{OUT}}$ reflects the 1000 frequency bins for each FRF. It contains identity activations since we use the metamodel to perform a multivariate regression.

The hyperparameters in Table 2 are chosen to reflect standard values within the state of the art [34–37]. A batch size of 10 and a total number of 200 epochs was combined with an Adam optimizer reducing a mean squared error (MSE) loss during training. For future

applications, the results could be further improved by a Bayesian optimization of the ANN hyperparameters.

**Table 2.** Hyperparameters for each trained ANN.

| Parameter | Value |
|---|---|
| Batch size | 10 |
| Epochs | 200 |
| Optimizer | Adam |
| Loss | MSE |
| Dropout $d_{HL1}$, $d_{HL2}$ | 20% |
| Dropout $d_{HL3}$ | 40% |

To prevent overfitting, a dropout of 20% is applied to the first two hidden layers and a dropout of 40% is applied to the third hidden layer. At the end of the number of epochs used for training, the validation loss lies under the training loss which validates regularization of the ANN [38]. The 45 ANNs were implemented in Tensorflow. Comparing the time consumption of both the ANN training and the training data generation (FE simulation), the data generation dominates by magnitudes. This confirms the necessity of an automated training data generation, achieved by the automated morphing algorithm.

### 2.4. Creation of the Target FRF

Usually, the target in NVH optimization is an amplitude reduction or a frequency shift. Simple mathematical functions representing maximum amplitudes for an FRF cannot demand frequency shifts, though. In our research, we wanted to provide design engineers with a fast and easy to use formulation of the optimization targets. In an intuitive process we define the target function by manipulating or drawing over the given initial FRF. This way, design engineers can define specific regions with amplitude changes, frequency shifts or clearance for deterioration. The drawing process is illustrated in Figure 3. The black line represents the initial FRF, the gray skyline in the background represents the same FRF for all 500 DoE parameter sets. The pictogram illustrates, which of the 45 FRFs is under investigation.

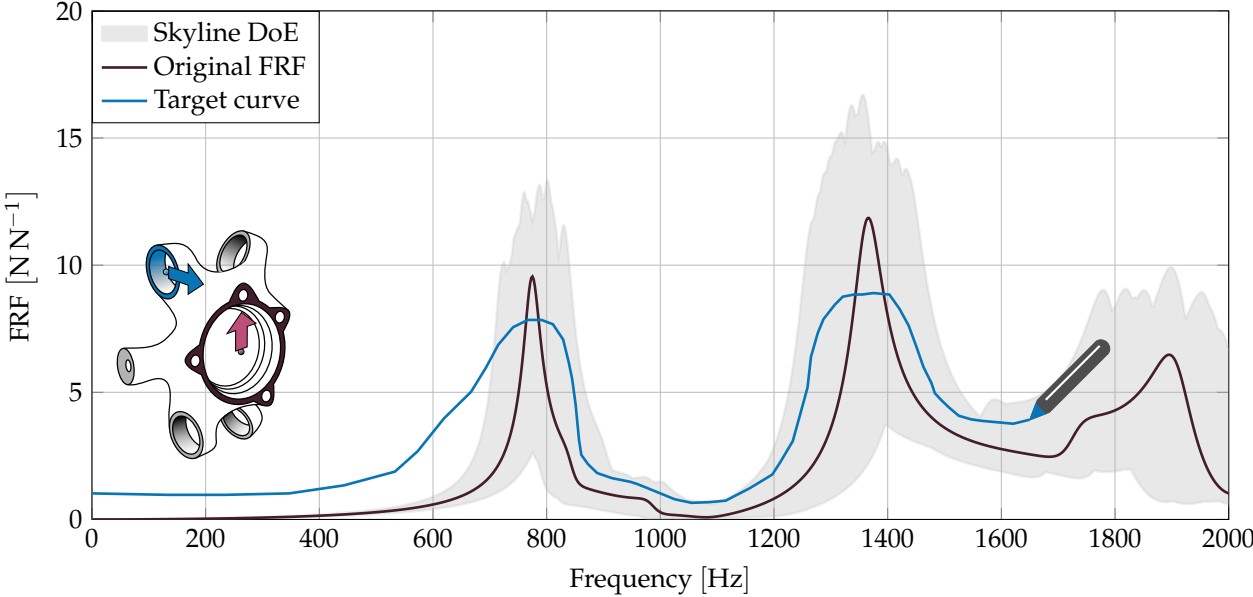

**Figure 3.** User input for the creation of a target curve. A graphical application presents the original FRF (black) and the skyline of all FRFs of the DoE data set (gray). The design engineer draws (pen symbol) a desired target curve into the plot (blue).

The target curve for the optimization can either act as a maximum-curve or a matching-curve. For the maximum-curve, the aim is to find a frequency response function that lies below the matching-curve. In contrast to that, with the specification of a matching-curve, the desired FRF has to show as little deviation as possible from the matching-curve.

### 2.5. Obtaining the New Parameter Set

For optimization, the EA implementation *NSGAII* provided in *Platypus* [39] is used under the *GNU General Public License 3* [40]. For each of the 45 FRFs, the optimizer uses one ANN from Section 2.3 to identify best fitting parameter sets to fulfill the target curve. The 15 modified geometry parameters of the part and the 1000 frequency bins of one FRF are used as input and output of each net, respectively.

The optimization criteria differ for a matching-curve and a maximum-curve. For the matching-curve, we use the sum of squared deviations

$$E_{\mathrm{matching}} = \sum_{j=1}^{1000} \left( A_{\mathrm{ANN},j} - A_{\mathrm{matching},j} \right)^2 \tag{4}$$

between the FRF amplitude estimation of the ANN $A_{\mathrm{ANN},j}$ and the matching-curve $A_{\mathrm{matching},j}$ for each frequency bin $f_j$ as minimization criterion $E$. This favors matching frequencies over matching amplitudes: The optimization criterion calculates the squared deviation for each frequency step. Thus, the deviation is larger for an inaccurate frequency compared to an inaccurate amplitude. For this reason the received designs tend to favour accurate frequency changes over accurate amplitude changes. This complies with common NVH demands for eigenfrequency shifting, which is used to avoid matching eigenfrequencies of multiple components in a transfer path [3,13,41].

For the maximum-curve, we calculate weighted differences between estimated and desired curve. The difference

$$E_{\mathrm{maximum}} = \sum_{j=1}^{1000} w_j \left( A_{\mathrm{ANN},j} - A_{\mathrm{maximum},j} \right) \tag{5}$$

is calculated by subtracting the amplitude values for each frequency bin. Estimated frequency bins above the maximum-curve are punished with a factor of $w_j = 10{,}000$. This is the same order of magnitude as the summed amplitudes of the whole spectrum. If the frequency bin amplitude lies below the maximum-curve, $w_j$ equals one. This punishment factor showed to be effective to prevent the optimizer from identifying a parameter set with an FRF which lies above the maximum-curve in any frequency bin. The sum of these weighted differences acts as the minimization criterion.

Both Equations (4) and (5) combine the deviation between target and design FRF for the whole frequency range into a scalar single-objective optimization criterion. In this way a drawn curve combining multiple amplitude and frequency changes is condensed into a single-objective optimization resulting in a single optimum design.

### 2.6. Process Verification

In order to verify the implementation of the ANNs and the optimization algorithm, we perform a test case, before going into the application examples section. For this test case, we pick one of the 45 FRFs. It is the FRF from Figure 3.

Since the dropout introduces randomness into the ANN training, the training outcome differs between multiple trainings, despite using the same parameters and input data. To verify a consistent successful training process, we monitor the MSE loss function for 100 individual trainings. The mean value of the MSE for the 100 trainings is $0.0744\,\mathrm{N}^2\,\mathrm{N}^{-2}$ with a standard deviation of $0.0065\,\mathrm{N}^2\,\mathrm{N}^{-2}$. This confirms a consistent successful ANN training.

For the selected FRF we use the FRF of the initial component design as a matching-curve. The initial design is not included in the 500 designs DoE data set, so the algorithm

has never seen the design of the target FRF. As we use a target curve for one of the 45 FRFs only, it would be coincidence, if the optimizer identifies the original component design, though. For each FRF there are only few significant design parameters. Therefore, we expect some design parameters to be determined with a small variance. Others—the non significant parameters for the selected FRF—with larger variance. Additionally, our experience shows that often there are multiple design parameter combinations to achieve a specific FRF. This complies with the aim of our approach to generate different component design suggestions for the design engineers, considering NVH demands.

In order to verify the optimization and the variance of the results, we perform 100 individual optimization runs with identical optimization parameters and target curve. As the *NSGAII* optimizer introduces a stochastic element, we expect the results to differ. Figure 4 presents the convergence plot for one of the optimization runs. For this run, the optimization criterion $E_{\text{matching}}$ (Equation (4)) is ultimately reduced to $20.11 \, \text{N}^2 \, \text{N}^{-2}$.

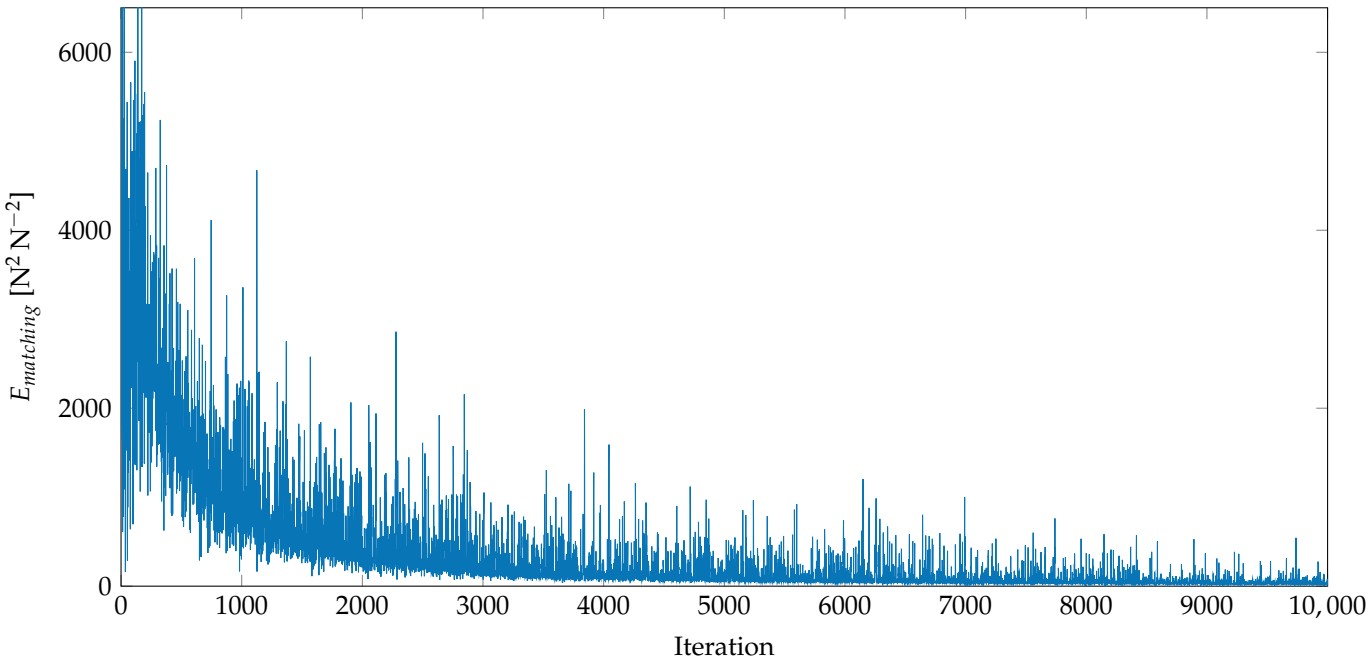

**Figure 4.** Optimization criterion $E_{\text{matching}}$ over optimization iterations.

For the 100 optimization runs, the mean value of the matching-curve optimization criterion is $18.07 \, \text{N}^2 \, \text{N}^{-2}$ with a standard deviation of $3.68 \, \text{N}^2 \, \text{N}^{-2}$. This confirms a consistent successful optimization process. As expected, the resulting optimum designs differ. The standard deviation of the 15 design variables differs between 1 mm and 5 mm. A small standard deviation around 1 mm indicates a significant design variable for the selected FRF. A large standard deviation indicates a design variables without any influence on the selected FRF. The optimizer chooses these design variables arbitrarily, as they do not influence the FRF.

The design variable with the lowest variance—and, thus, the one with the largest influence on the FRF—has a standard deviation of 1.21 mm. This design variable is the $x_1$-coordinate of the blue hard point in Figure 3. This design variable determines dominantly the lever arm between the wheel center point and the kinematic hard point. This shows the optimizer's capability to identify relevant design parameters. The presented result concurs to the hypothesis from an earlier publication [14]. Small changes to the location of few kinematic hard points can influence the noise transfer while others can be used for different development domains.

## 3. Application Examples

In this section we present and discuss practical results and the capabilities of the approach using the scenario, described in Section 2.1. We present different applications—typical for NVH development—using a target curve for one of the 45 FRFs for each application. The applications include *amplitude reductions* (Sections 3.1 and 3.2), *changes of eigenfrequencies* (Section 3.3), and *combined optimizations* (Section 3.4). For the application examples, we use different FRFs in order to showcase the general applicability of aimed FRF modifications for multiple different use cases.

As the approach is aimed at the early digital design phase, we want to generate designs with aimed changes in the FRFs. An exact match between prediction and validation is desired, but not mandatory for the design process.

In the following graphs, the FRF of the initial component is highlighted by a black line. The gray skyline in the background visualizes the minimum and maximum variation of the corresponding FRF values for all of the 500 DoE parameter sets. The blue curve marks the optimization target curve, either matching-curve or maximum-curve. The predicted FRF resulting from the ANN is highlighted in red. The simulated validation FRF is highlighted in green. For an ideal ANN, the predicted and simulated curves match. The FRF amplitudes are plotted logarithmically so that small differences are easier to recognize.

### 3.1. Target FRF with Amplitude Reduction

In a first investigation we showcase an amplitude modification. We use the original FRF as a target curve, differing only at the 1700 Hz peak. For this peak, we demand an amplitude reduction. The received FRF should lie on this matching-curve.

In Figure 5, the blue matching-curve flattens the peak. The predicted and simulated FRFs clearly show a reduction in the amplitude in the demanded frequency range.

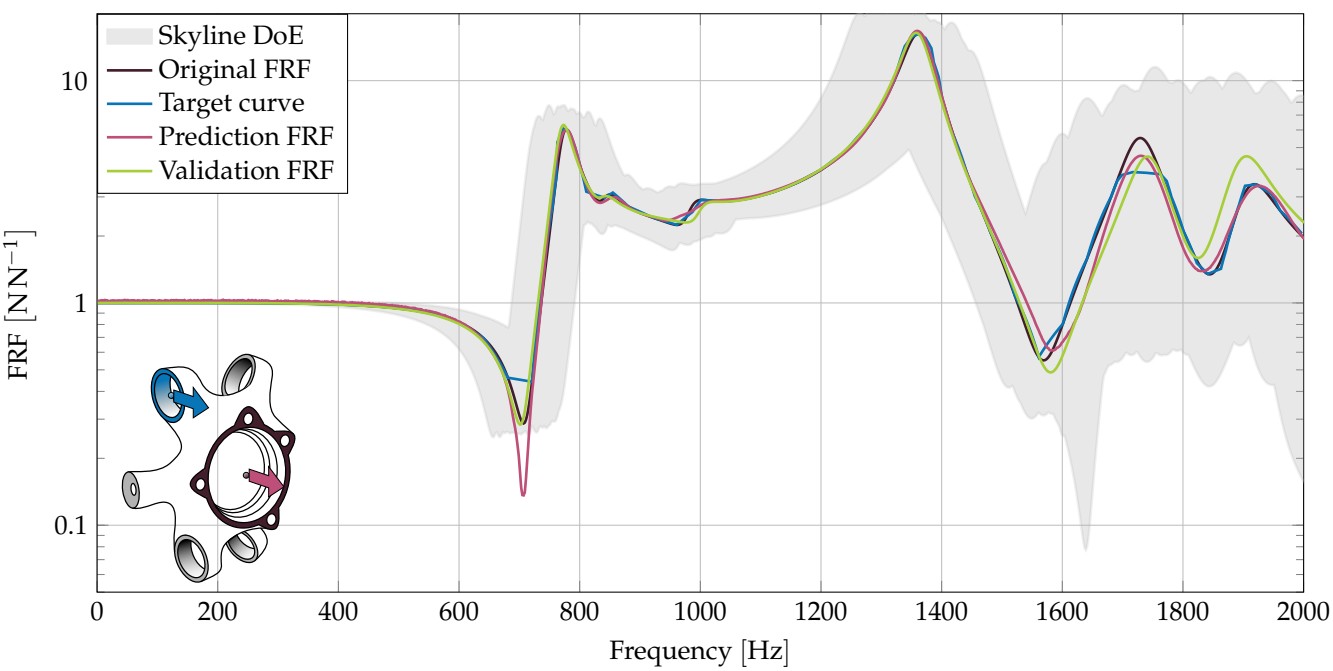

**Figure 5.** Optimization using a matching-curve for a desired amplitude reduction around 1700 Hz.

At 1900 Hz we clearly see the difference between estimated and simulated FRF. This indicates a minor ANN estimation quality for the frequency bins above 1800 Hz, compared to lower frequency ranges.

The skyline implies a much larger potential for improvement around 1700 Hz. This improvement would be accompanied by large deviations in other frequency ranges, though.

This is prohibited by the matching-curve. The example shows the possibility to perform aimed improvement without deterioration in other areas.

### 3.2. Maximum-Curve with Multiple Amplitude Optimization Targets

In the next example we showcase the possibility to draw maximum-curves describing a multi-objective optimization. The blue maximum-curve in Figure 6 is the one, drawn in Figure 3. The aim is the amplitude reduction at all three peaks. In contrast to the previous example, the optimizer receives clearance around the peaks to shift them, as long as the amplitude is reduced.

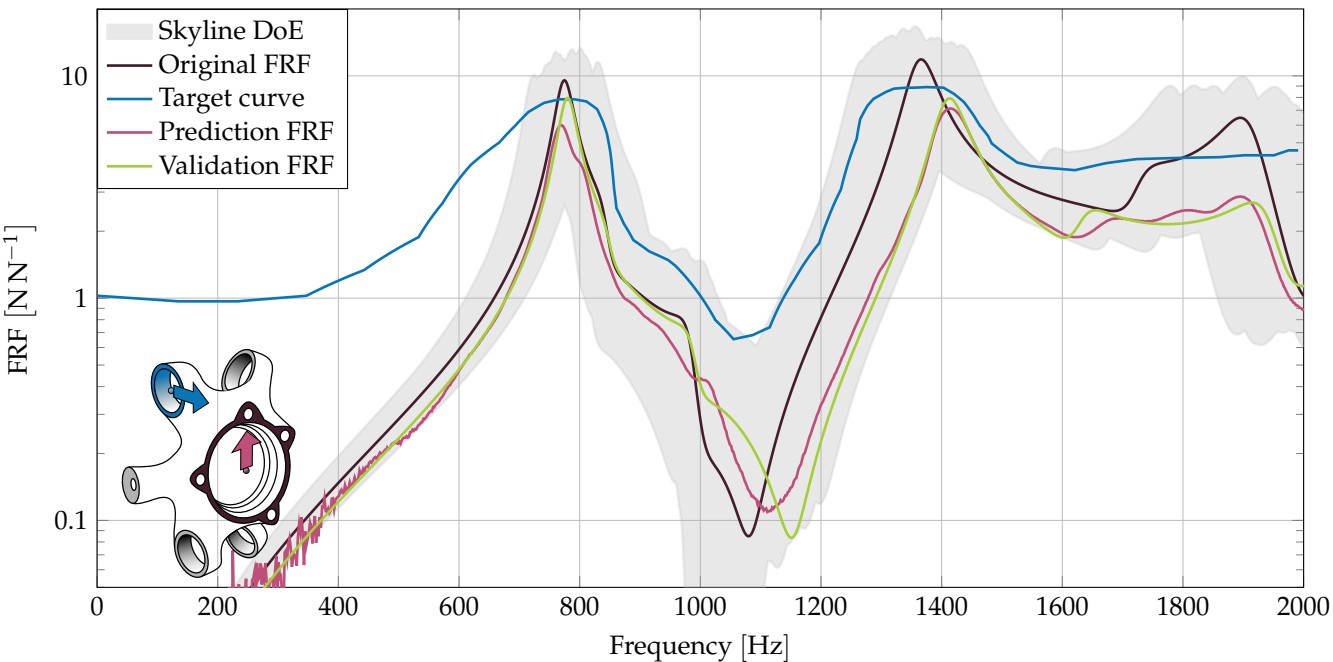

**Figure 6.** Optimization using a drawn maximum-curve describing a multi-objective optimization. Desired amplitude reduction at multiple frequency ranges while allowing small eigenfrequency shifts.

The multiple amplitude reduction targets at the three FRF peaks are combined into a single objective optimization using Equation (5). Therefore, the optimization result is a single design.

It is clearly visible that the optimizer is able to identify a parameter set that fulfills the multi-objective optimization demanded by the drawn maximum-curve. The clearance at 1400 Hz enables an eigenfrequency shift to achieve the desired amplitude reduction. Again, the example shows that the prediction of the amplitudes at eigenfrequencies with the ANN could receive further improvement. In general, the ANN does not predict eigenfrequencies, but amplitude changes for each frequency bin. The behavior shows a higher prediction quality for frequency bins with minor changes and lower quality for frequency bins with larger changes, i.e., frequency ranges containing eigenfrequencies. These prediction quality differences combined with the selected minimization criterion, which favors matching frequencies over matching amplitudes, makes the approach especially suitable for desired frequency shifts.

### 3.3. Change of Eigenfrequency

As the previous example hinted a suitability for eigenfrequency shifts, we demand an aimed eigenfrequency change without the demand of an amplitude reduction, next. This resembles a common demand in NVH development. In Figure 7 we use the same FRF as used in Figure 5. We define a matching-curve demanding an eigenfrequency raise at 1700 Hz, without modifying the remaining FRF.

The new parameter set raises the eigenfrequency by approximately 50 Hz, which is half the desired raise of 100 Hz. Again, frequency prediction is more accurate, compared to amplitude prediction.

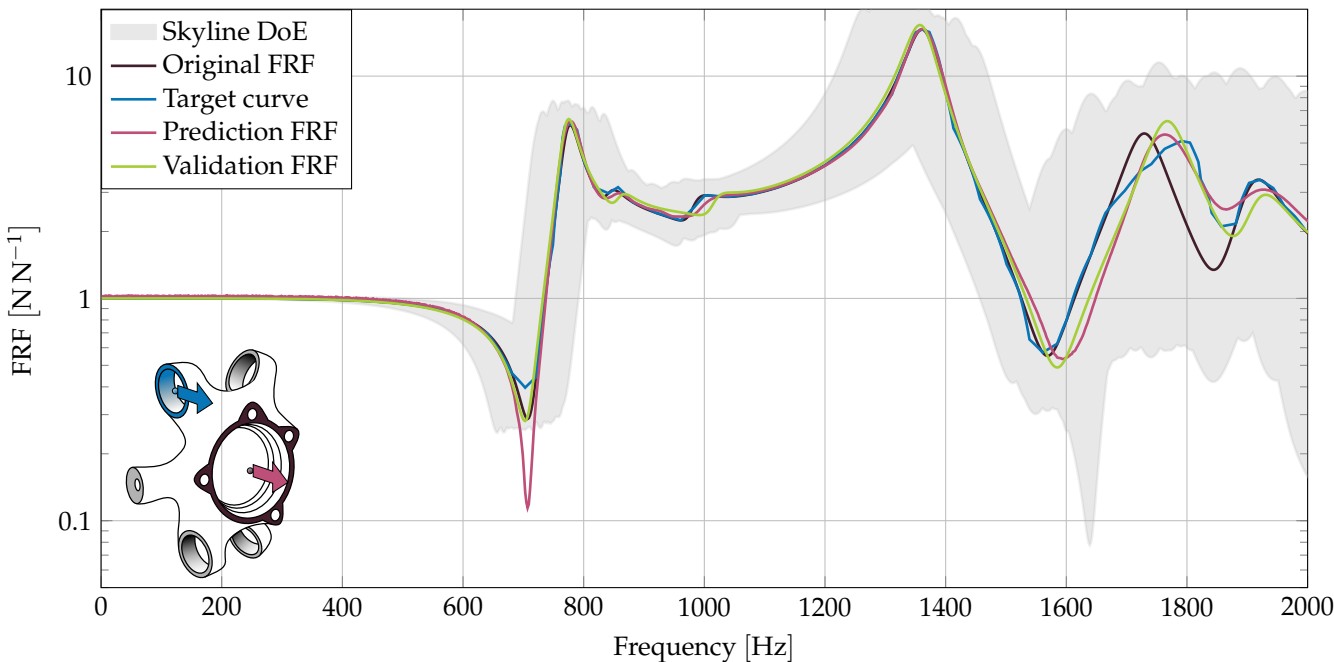

**Figure 7.** Optimization using a matching-curve to achieve an eigenfrequency raise at 1700 Hz.

We demanded an eigenfrequency reduction for another FRF, to showcase both directions of change. In Figure 8 we reduce the eigenfrequency at 1900 Hz without an amplitude reduction target. In this case, the target frequency is matched accurately.

Both examples show the possibility to move specific component eigenfrequencies. The target curves not only consider the eigenfrequency itself, but also the rising and falling area left and right of the actual eigenfrequency. This width would not be considered by a scalar eigenfrequency optimization.

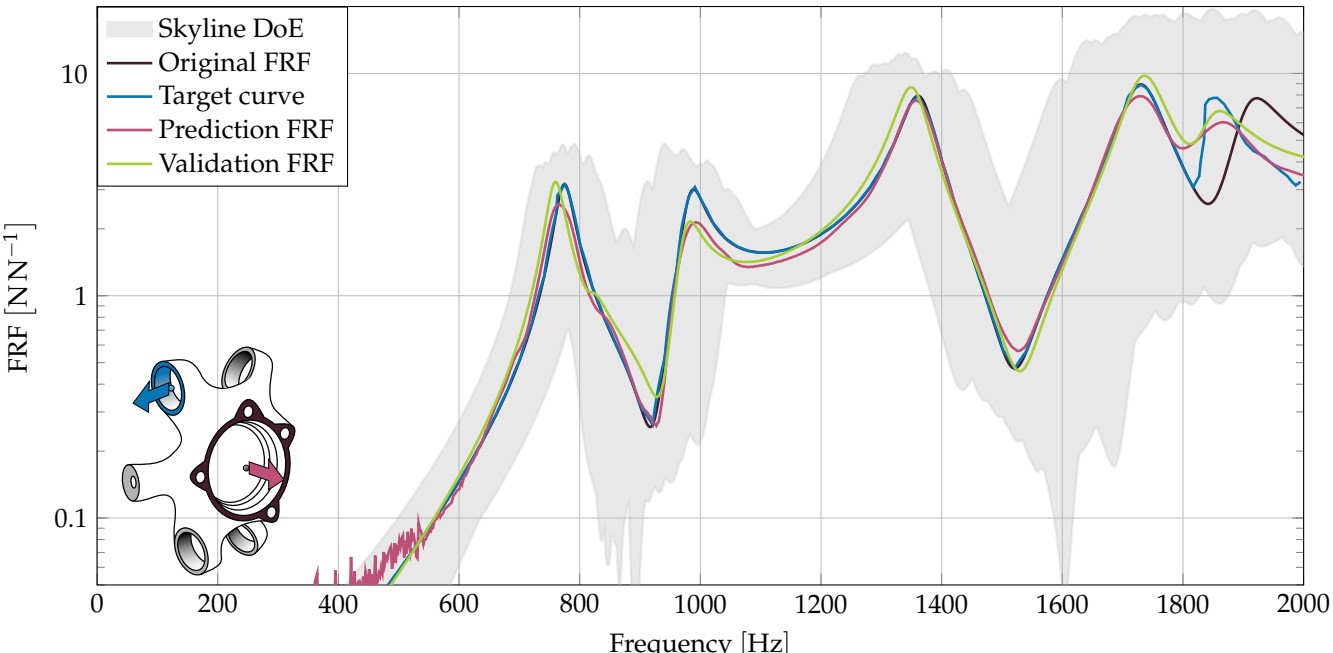

**Figure 8.** Optimization using a matching-curve to achieve an eigenfrequency reduction at 1900 Hz.

### 3.4. Combined Amplitude and Frequency Changes

The following three examples try to combine both frequency and amplitude changes. For Figures 9 and 10 we used a matching-curve and tried to alter one eigenfrequency, without any changes to the remaining FRF. Both examples show a suitable frequency shift without any amplitude improvements.

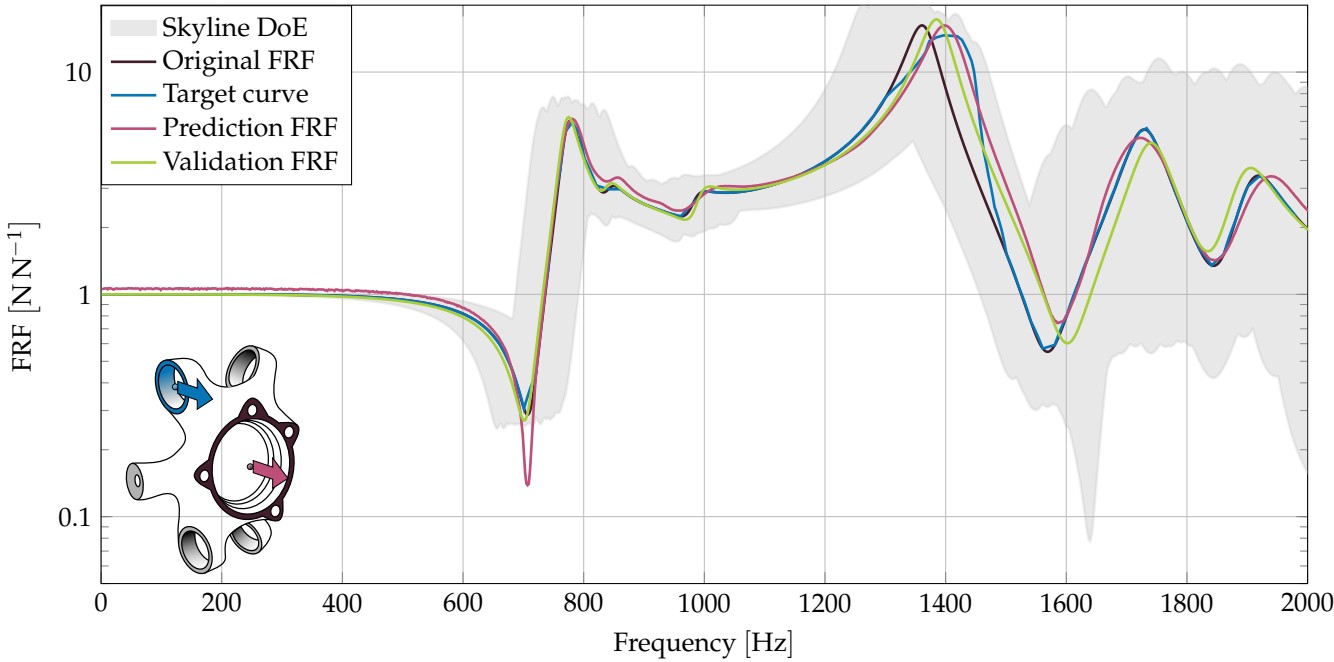

**Figure 9.** Optimization using a matching-curve to achieve an eigenfrequency raise and an amplitude reduction at 1350 Hz.

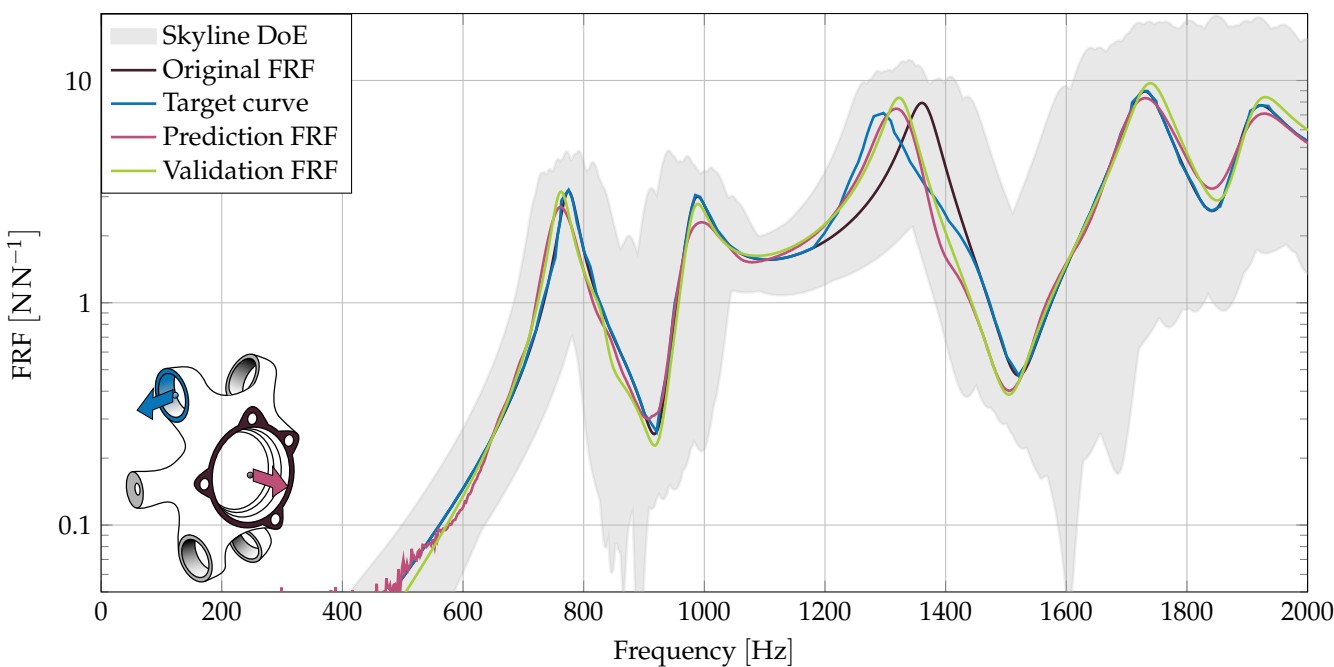

**Figure 10.** Optimization using a matching-curve to achieve an eigenfrequency reduction and an amplitude reduction at 1350 Hz.

For Figure 11 we used a maximum-curve and granted more freedom in changing the FRF. This additional freedom enabled the optimizer to find a parameter set that both

fulfilled the frequency reduction and the amplitude reduction. The less strict boundary condition even enabled improvement in additional frequency ranges.

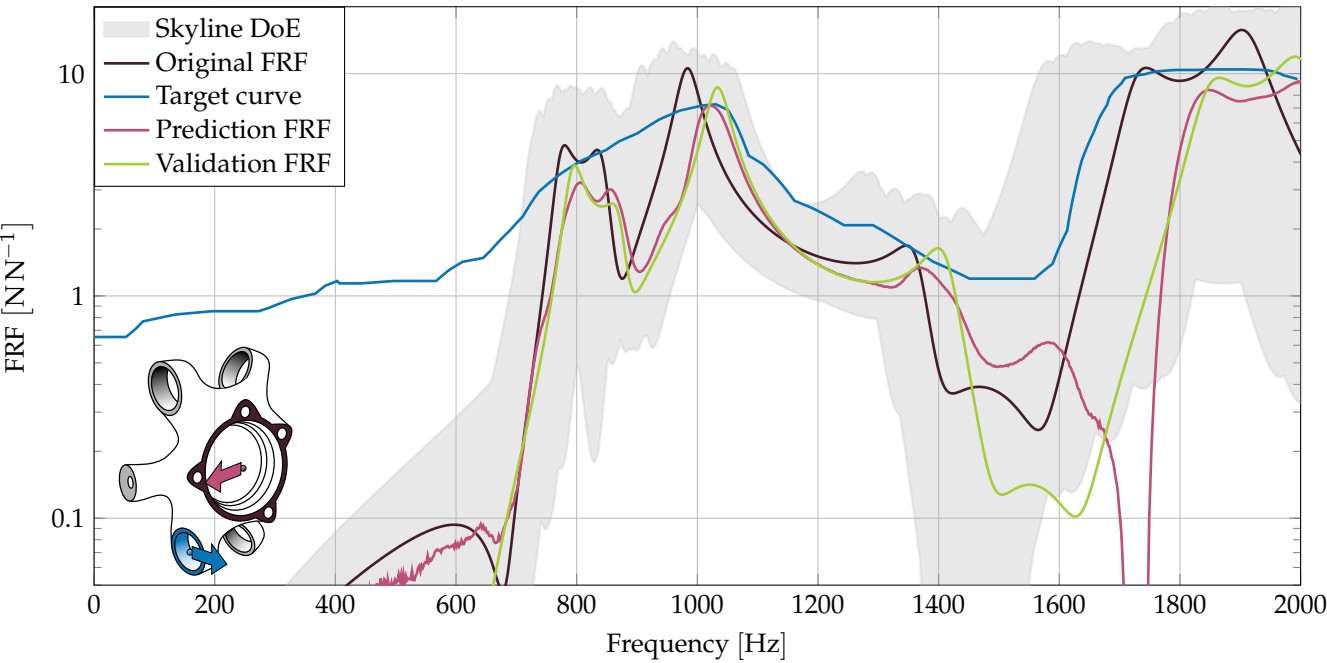

**Figure 11.** Optimization using a maximum-curve to achieve an eigenfrequency and amplitude reduction at multiple frequency ranges.

### 3.5. Comparison to Earlier Investigations

In an earlier publication, we used polynomial metamodels for a similar optimization problem [27]. There, polynomial metamodels performed well for the prediction of a single frequency bin with only three design variables. This approach had to be expanded for the use with more design variables and more complex systems.

The presented application examples provide us with insights into the suitability of ANNs as metamodels for the NVH suspension development. For us, the most important finding is that the required ANN training time is magnitudes smaller compared to the necessary NVH FE simulation for the DoE designs. For this reason, longer training time for the metamodel is not an argument against using ANNs. This enables us to use all the advantages of ANNs for the representation of more complex systems. This will be especially important for the desired expansion to sub-assembly or even full vehicle simulations. First investigations into these expansions hint complex dependencies that cannot be represented by polynomial metamodels with acceptable numbers of DoE samples.

## 4. Conclusions and Outlook

In this article we presented an approach to use ANNs as metamodels for an NVH FE simulation in an optimization workflow. For a vehicle suspension component, we determined geometry parameter sets, with which the component FRF meets a target curve. On the one hand, we presented the possibility to modify an FRF in a specific frequency range only, while preserving the rest of the FRF. On the other hand we freed the design space giving the optimizer more possibilities to alter the FRF. Both cases proved the possibility to either change the frequency of specific FRF features or their amplitude. In a last example we proved a combination of frequency and amplitude change possible, if the design space is open enough. Less strict target curves tend to provide better component designs.

The presented examples showed the possibility to create designs satisfying the given FRF restrictions. This can support the component design process in the early digital development phase. Giving rough restrictions to the FRFs can hint new design options.

Narrow restrictions can lead to an optimized component performance without changing sufficient areas.

The presented examples confirmed the possibility to use ANNs as metamodels in a conventional optimization process for aimed changes in FRFs. Comparing estimated FRFs to simulated FRFs hinted a higher prediction quality for eigenfrequency position compared to eigenfrequency amplitude. The prediction quality could be further improved by optimizing the ANN hyperparameters.

The investigation shows potential for aimed optimization of the NVH performance. In future work, the investigation of component FRFs could be expanded onto sub-assembly FRFs or even full vehicle simulations. Then, the optimization criteria could not only include NVH characteristics, but also characteristic values for other development domains.

**Author Contributions:** Conceptualization, T.v.W. and F.G.; methodology, T.v.W., F.R. and D.E.T.; software, T.v.W., F.R. and D.E.T.; validation, T.v.W.; writing—original draft preparation, T.v.W., F.R. and D.E.T.; writing—review and editing, T.v.W., F.R., D.E.T. and F.G.; visualization, T.v.W.; supervision, F.G.; project administration, T.v.W. All authors have read and agreed to the published version of the manuscript.

**Funding:** This research was funded by Mercedes-Benz AG.

**Data Availability Statement:** Data sharing is not applicable to this article.

**Acknowledgments:** We thank the Mercedes-Benz AG with the department NVH road noise, especially Ernst Prescha, Stéphanie Anthoine, Christian Olfens, and Wibke Lommatzsch for their support. Additionally, we would like to thank the members from the team and department for always being available to discuss the ongoing research. We thank Achim Winandi from the Karlsruhe Institute of Technology for valuable input and support. We acknowledge support by the KIT-Publication Fund of the Karlsruhe Institute of Technology.

**Conflicts of Interest:** The authors declare no conflict of interest. Timo von Wysocki, Frank Rieger, and Dimitrios Ernst Tsokaktsidis are from Mercedes-Benz AG, the company had no role in the design of the study; in the collection, analyses, or interpretation of data; in the writing of the manuscript, and in the decision to publish the results.

## Abbreviations

The following abbreviations are used in this manuscript:

| | |
|---|---|
| ANN | Artificial Neural Network |
| FE | Finite Element |
| FEM | Finite Element Method |
| FRF | Frequency Response Function |
| HL | Hidden Layer |
| MSE | Mean Squared Error |
| NVH | Noise, Vibration, and Harshness |
| ReLu | Rectified Linear Unit |
| SFLH | Space Filling Latin Hypercube |

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
