# Peer review of "Generating Component Designs for an Improved NVH Performance by Using an Artificial Neural Network as an Optimization Metamodel"

_designs_

Round 1
Reviewer 1 Report
The article presents a design method to improve frequency responses of a vehicle component, based on an ANN used as a metamodel to map geometry parameters and the frequency response. The study is well motivated, with a clear explanation of the unique contribution of the current manuscript compared to related works by the authors. The choices of inputs and outputs to the neural net model are well-reasoned, and the examples are effective at highlighting the insight into what optimization problems are likely to lead to desired outcomes. The article is generally written very well, but further explanations of the training process of the ANN would help to understand the cost of data generation/training and the reliability of the ANN model. Please consider the following suggestions to address this concern.
- The data size of 500 seems fairly small considering the input and output dimensions. This may be chosen as a trade-off between modeling accuracy vs computational cost of generating the training data, however, not much explanation on this point is offered. Any descriptions of how these considerations play into this choice would be helpful.
- Reporting on how well the ANN model is expected to predict the frequency response should be included. This could include the result of the ANN training process or reporting the prediction accuracy for randomly selected designs.
Author Response
Dear Reviewer,
Thank you very much for your valuable feedback. It helped us a lot to improve the quality of our manuscript.
Thank you for revealing the lack of information concerning the data size. Indeed, the 500 samples are a trade-off between simulation time and metamodel accuracy. Following your hint, we added a paragraph in section 2.2, clarifying our choice.
We added section 2.6 containing metrics regarding the ANN training process as well as the optimization process. We performed a case study with multiple training and optimization runs and present the characteristic numbers quantizing the performance and prediction accuracy of our approach.
Best regards
Timo von Wysocki
Reviewer 2 Report
The authors present the optimization of the frequency response function of components by ANN trained via FE analysis.
The paper is well written but needs to be revised. In the following the major comments.
1 - I was not familiar with the concept of NHV, so I think that in both title and abstract it should be written without the acronym
2 - The definition of the 5 hard point for the design is not clear. Also, the 15 design variables are not well explained. Maybe some further figures and/or equations are needed to describe the design modification.
3 - Have been performed grid verification analysis? If yes, the authors should also provide the grid sensitivity study. Furthermore, I would suggest adding a figure of the FE model/grid.
4 - The NSGA II is stochastic in nature, so statistically significant results can be achieved only with optimization repetitions. Since the optimization is performed on the ANN as a metamodel, it is costless, and the authors should perform something like 100 optimization repetitions, providing average results with confidence interval (defined by quantiles or standard deviations)
5 - The hyperparameters of the ANN have been fixed a priori, I would suggest the use of Bayesian optimization for their optimal definition.
6 - In the results, is not clear what the skyline of the DoE is. Does it represent the maximum variation among all the 500 samples of the DoE? If yes, why it is not constant for all the problems solved, since I understand that the design space is always the same, but the only modification is the objective function.
7 - Convergence plot and Pareto front should be shown. Furthermore for the multi-objective problems is not clear what and how many are the objectives.
8 - Since the authors have used dropout in the ANN training, which helps to generalize, they should repeat the training a number of times, since dropout is a stochastic procedure, that leads to different results each time it is used. SO the results of an ANN should be represented by the expected value and standard deviation/variance of the net
Author Response
Dear Reviewer,
Thank you very much for your detailed feedback. It was valuable to improve the quality of our manuscript. Please find below our point by point improvements, following your suggestions:
- In the NVH-community, the non-acronym form "Noise, Vibration, and Harshness" is very uncommon. Therefore we would like to keep the acronym in the title to increase visibility in the community. We fully agree though, that in the abstract we should introduced the acronym.
- Thank you very much for pointing out the unclear description of the hard points and design variables. Following your suggestion, we completely reworked Figure 1 and modified the description text. We hope this clarifies the study setup. Additionally we added a visualization to each plot of the results section in order to clarify which FRF is under investigation.
- In section 2.2 we added a new paragraph presenting the results of an FE mesh verification study. Here, we investigated the mesh quality after the morphing process for a worst case scenario. The results indicate no loss in mesh quality. Regarding a component figure, we detailed Figure 1 for a clearer component understanding. Additionally we added a hint to a previous publication including a figure of the FE mesh, as the mesh was in focus there.
- Thank you for suggesting a systematic investigation of the NSGA-II optimization. Performing 100 optimization runs provided meaningful insights into the optimization process: We added section 2.6 containing an investigation of the optimization reproducibility. We provide average results with standard deviations for both the optimization criterion and the resulting design parameter optima. Additionally, we discuss the underlying physical principle for the presented optimization result.
- Thank you for the valuable suggestion on how to further improve our results. We fully consent to the potentials of a Bayesian hyperparameter optimization. As our article focuses first on providing a general design approach valuable for the NVH-community and second on a best performing ANN, we would like to reserve this optimization for the future work, as it would not be possible in the revision time. We added these points to the manuscript.
- We added a paragraph to the beginning of section 3 containing an improved description of the used FRFs and the skyline. Indeed, the design space is always the same, but there are different FRFs, as we want to demonstrate a general usability for different use cases. Additionally, we added a pictogram to each plot in order to clarify the FRF under investigation
- We added paragraphs to section 2.5 and 3.2 explaining how the multi-objective optimization is converted into a single objective optimization using equation 5. We hope this clarifies that there is no pareto front but a single optimum design. Additionally we added Section 2.6 investigating the optimization process. We present characteristic values for the optimization reproducibility and present-as suggested-a convergence plot for an optimization example.
- We added a new section 2.6 in which we added multiple metrics evaluating the ANN training and optimization methods. For the network training we performed 100 training runs for one FRF and present the differences in MSE via mean value and standard deviation.
Thank you again for your valuable feedback, which improved our manuscript heavily.
Best regards
Timo von Wysocki
Round 2
Reviewer 2 Report
The authors have addressed the main comments and the paper can be accepted